

# Effects of complex training versus heavy resistance training on neuromuscular adaptation, running economy and 5-km performance in well-trained distance runners

Fei Li[1], Ran Wang[1], Robert U. Newton[2], David Sutton[3], Yue Shi[1] and Haiyong Ding[1]

[1] School of Physical Education and Sport Training, Shanghai University of Sport, Shanghai, China
[2] School of Medical and Health Sciences, Edith Cowan University, Joondalup, Australia
[3] Talent identification Center and Research Institute, Shanghai Sports School, Shanghai, China

## ABSTRACT

**Background.** Recently, much attention has been paid to the role of neuromuscular function in long-distance running performance. Complex Training (CT) is a combination training method that alternates between performing heavy resistance exercises and plyometric exercises within one single session, resulting in great improvement in neuromuscular adaptation. The purpose of this study was to compare the effect of CT vs. heavy resistance training (HRT) on strength and power indicators, running economy (RE), and 5-km performance in well-trained male distance runners.

**Methods.** Twenty-eight well-trained male distance runners (19–23 years old, $VO_{2max}$:65.78 $\pm$ 4.99 ml.kg$^{-1}$.min$^{-1}$) performed one pre-test consisting of: maximum strength (1RM), counter movement jump (CMJ) height, peak power, a drop jump (DJ), and RE assessments, and blood lactate concentration (BLa) measurement at the speeds from 12–16 km.h$^{-1}$, a 50-m sprint, and a 5-km running performance test. They were then divided into 3 groups: complex training group (CT, $n = 10$), that performed complex training and endurance training; heavy resistance training group (HRT, $n = 9$) that performed heavy strength training and endurance training; and control group (CON, $n = 9$) that performed strength-endurance training and endurance training. After the 8 weeks training intervention, all participants completed a post-test to investigate the training effects on the parameters measured.

**Results.** After training intervention, both the CT and HRT groups had improvements in: 1RM strength (16.88%, $p < 0.001$; 18.80%, $p < 0.001$, respectively), CMJ height (11.28%, $p < 0.001$; 8.96%, $p < 0.001$, respectively), 14 km.h$^{-1}$ RE (−7.68%, $p < 0.001$; −4.89%, $p = 0.009$, respectively), 50-m sprints (−2.26%, $p = 0.003$; −2.14%, $p = 0.007$, respectively) and 5-km running performance (−2.80%, $p < 0.001$; −2.09%, $p < 0.001$, respectively). The CON group did not show these improvements. All three training groups showed improvement in the 12 km.h$^{-1}$ RE ($p \leq 0.01$). Only the CT group exhibited increases in DJ height (12.94%, $p < 0.001$), reactive strength index (19.99%, $p < 0.001$), 16 km.h$^{-1}$ RE (−7.38%, $p < 0.001$), and a reduction of BLa concentrations at the speed of 16 km.h$^{-1}$ (−40.80%, $p < 0.001$) between pre- and post-tests.

Corresponding author
Haiyong Ding,
dinghaiyong@sus.edu.cn

**Conclusion**. This study demonstrated that CT can enhance 1RM strength, CMJ height, 12 and 14 km.h$^{-1}$REs, 50-m sprints and 5-km running performances in well-trained male distance runners and may be superior to HRT for the development of reactive strength and 16 km.h$^{-1}$RE, and reduction of BLa concentrations at speed of 16 km.h$^{-1}$. Young male distance runners could integrate CT into their programs to improve the running performance.

# INTRODUCTION

Maximum oxygen uptake (VO$_{2max}$), lactate threshold (LT), and running economy (RE) have been determined to be the most crucial physiological factors influencing long-distance running performance (*Midgley, McNaughton & Jones, 2007*). As running pace increases (*Díaz, Fernández-Ozcorta & Santos-Concejero, 2018*), runners must be able to sustain a relatively high speed over the course of competition, during which the practical energy requirements may surpass the power output of their aerobic system (*Paavolainen, Nummela & Rusko, 2000*). Because elite runners share similar levels of VO$_{2max}$ and LT (*Beattie et al., 2014*), it is difficult to use these factors to distinguish the endurance performance. Consequently, in addition to central factors such as cardiovascular capacity, peripheral factors relating to neuromuscular function also play a critical role in endurance performance (*Nummela et al., 2006*; *Paavolainen, Nummela & Rusko, 2000*). In trained endurance runners, chronic strength training elicits increases in RE coincident with improved performance, that is partially due to neuromuscular adaptations associated with strength training (*Beattie et al., 2017*; *Piacentini et al., 2013*). Heavy resistance training (HRT) and plyometric training (PLY) are the most common strength training methods utilized by distance runners due to their role in developing the muscle force production and function of the stretch-shortening cycle (SSC), which are critical contributors for distance running (*Spurrs, Murphy & Watsford, 2003*; *Storen et al., 2008*). However, insufficient research has been conducted on the impact of simultaneous heavy strength and plyometric training on endurance performance.

Complex training (CT) is described as a combination training method that alternates between performing heavy resistance exercises and plyometric exercises within one single session (*Macdonald, Lamont & Garner, 2012*). These two types of exercises performed consecutively in CT are referred to as a 'complex pair' that has significant effects on the development of strength and power (one repetition maximum strength [1RM], jump, and sprint performance) in team-sport athletes (*Maio Alves et al., 2010*; *Santos & Janeira, 2008*). CT can elicit a post-activation potentiation (PAP) response, thus allowing individuals to produce more power on the subsequent exercise (*Carter & Jeremy, 2014*). The specific mechanisms include stimulating high order motor unit recruitment and excitability, increasing phosphorylation of the myosin light chain, and changes in limb

stiffness (*Blagrove, Howatson & Hayes, 2018*; *Tillin & Bishop, 2009*). These neuromuscular adaptations may also benefit long-distance performance such as improvement in RE and maximal running speed (*Beattie et al., 2014*; *Blagrove, Howatson & Hayes, 2018*). In addition, CT is a time-efficient training method allowing distance runners who have limited time for strength training to perform two types of training at the same time. Although CT has consistently been shown to improve performance for team-sport athletes, to the best of our knowledge, the benefit of CT in terms of neuromuscular adaptation, RE and endurance performance has received far less attention. Understanding the role of CT has a practical application for guiding distance runners' training.

Therefore, the primary purpose of this study was to investigate the effect of an 8-week CT regimen on neuromuscular adaptation (1RM strength, jumping performance, and reactive strength), running economy, and 5-km running performance in well-trained male runners. As previous research has already confirmed the impact of HRT on endurance performance (*Piacentini et al., 2013*; *Storen et al., 2008*), this study also aimed to compare the effect of CT and HRT. We hypothesized that CT performed concurrently with endurance training will lead to greater neuromuscular adaptations and endurance performance improvements than HRT in well-trained male long-distance runners.

## METHODS

### Experimental approach to the problem

The current study used a between-group repeated-measure design to test the hypothesis that CT improves neuromuscular adaptations, running economy, and endurance running performance to a greater extent as compared to that with HRT. This study began in the early stage of the winter training block (November, December, and January), lasted for 10 weeks, and was comprised of four parts: 2 weeks of preparatory strength training, a pre-test, 8 weeks of training intervention, and a post-test (Fig. 1). After the preparatory strength training, all participants underwent a two-day test protocol to assess body composition, 1RM strength, counter movement jump (CMJ), and drop jump (DJ), running economy, $VO_{2max}$, and 50-m and 5-km running performance as the pre-test. They were then divided into three groups matched by age, body composition, $VO_{2max}$, and 5-km running time. Groups consisted of: a complex training group, CT ($n = 10$) who performed complex training and endurance training; a heavy resistance training group, HRT ($n = 9$) who performed heavy strength training and endurance training; and a control group, CON ($n = 9$) who performed strength-endurance training and endurance training. After the 8-week training intervention, all participants completed a post-test, which was identical to the pre-test.

### Participants

Twenty-eight male runners on a collegiate long-distance running team volunteered to participate in this study and their basic physical characteristics, related body composition, and physiological indicators are provided in Table 1. All the participants competed at the collegiate level, they had a minimum of four years of training experience for long-distance running and were free from injury. They stayed and trained together at the

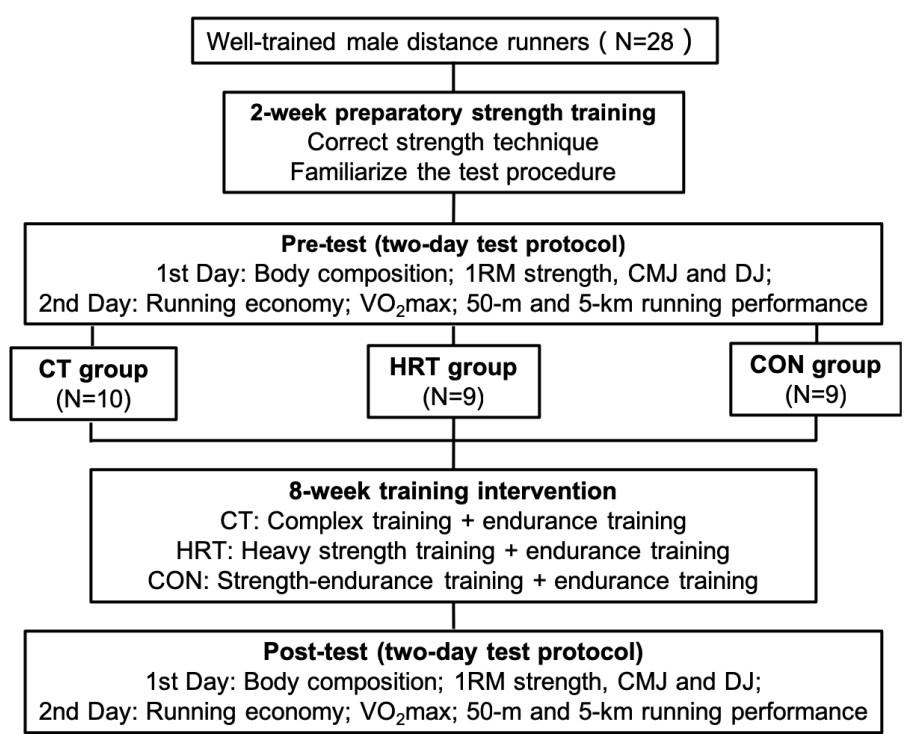

**Figure 1** Flowchart of the study design.

**Table 1** Baseline characteristics of subjects in each training group (Mean ± SD).

| | Complex training group ($N = 10$) | Heavy resistance group ($N = 9$) | Control group ($N = 9$) | F value | p value |
|---|---|---|---|---|---|
| Age (years) | 20.2 ± 1.03 | 21.22 ± 1.48 | 20.78 ± 1.20 | 1.616 | 0.219 |
| Height (cm) | 178.2 ± 6.1 | 175.3 ± 3.8 | 178.5 ± 4.4 | 1.146 | 0.334 |
| BM (kg) | 63.08 ± 6.08 | 57.76 ± 3.70 | 61.36 ± 4.49 | 2.863 | 0.076 |
| BMI (km.m$^{-2}$) | 19.86 ± 1.61 | 18.77 ± 1.03 | 19.24 ± 0.81 | 1.850 | 0.178 |
| FFM (kg) | 55.99 ± 4.86 | 52.26 ± 3.04 | 55.97 ± 4.02 | 2.545 | 0.099 |
| FM (kg) | 7.09 ± 3.14 | 5.50 ± 1.99 | 5.39 ± 1.70 | 1.516 | 0.239 |
| VO$_{2max}$ (ml.kg$^{-1}$.min$^{-1}$) | 65.65 ± 5.06 | 65.54 ± 5.52 | 66.14 ± 5.25 | 0.033 | 0.968 |
| 5-km time (s) | 953.70 ± 12.30 | 952.56 ± 10.10 | 954.11 ± 6.75 | 0.058 | 0.944 |

**Notes.**

BM, body mass; BMI, body mass index; FFM, fat-free mass; FM, fat mass; VO$_2$ max, maximum oxygen uptake.

training center affiliated with the Shanghai University of Sport, China, thus having similar training, recovery, and nutritional environments. The participants and their coaches were extensively informed about the research process and potential risks associated with the study and subsequently signed an informed consent document before participation. All participants were free from any health problems that would affect physical performance or put them at risk. The study was approved by the Ethics Committee of Shanghai University of Sport, China (ID number:2017047).

## Procedures
### Preparatory strength training

During the preparatory strength training phase, all participants were asked to master the correct strength training techniques and familiarize themselves with the testing procedures to ensure training efficacy and testing accuracy. All exercises employed during the 2-week preparatory strength training phase were the same as that in the training intervention except that the load was reduced to a minimum (all participants used an empty 20 kg barbell). All three training groups performed their routine endurance training during this phase.

### Pre-test and post-test

All participants performed a two-day test protocol. They were tested for by body composition, 1RM strength, CMJ, and DJ on the first day, and performed RE, $VO_{2max}$, 50-m and 5-km running performance tests on the second day. Participants fasted two hours before the test, and wore the same running shoes during the pre- and post-test.

### Body composition test

Height was measured using a wall-mounted stadiometer (Butterfly, Shanghai, China) and recorded to the nearest 0.1 cm. Body mass, fat mass and fat-free mass were measured using a bioimpedance analyzer (X-scan plus II, Jawon, Daejeon, South Korea). Body mass index (BMI) was calculated as body mass divided by height squared.

### Neuromuscular adaptation tests

As muscle force and power production and the function of the SSC are important factors for distance runners' performance (*Spurrs, Murphy & Watsford, 2003*; *Storen et al., 2008*), neuromuscular adaptation was tested using the 1RM strength test, CMJ and DJ test.

1RM strength test. This test was performed in a power rack (Hammer Strength, Rosemont, IL, USA) using a previously described protocol for measuring 1RM back squat strength (*MCBride et al., 1999*). Following the general warm up, each subject started with four warm-up sets of 10 reps at 50%, five reps at 70%, three reps at 80% and one rep at 90% of their estimated 1RM. Each subject's warm-up 1RM load was estimated by the researchers based on their body weight, training experience and age. After the warm-up sets, the subject performed three to four-effort trials to determine their actual 1RM, with a rest interval of 3 to 5 mins. Each subject was asked to squat through a full range of motion by lowering the bar to the point where the thigh was parallel with the ground. Two spotters monitored the participants throughout the test.

Counter movement jump. After the 1RM test, each participant rested for 15 mins, and then performed a CMJ and DJ test from a 40 cm height. Jumping height, peak power (maximum power during CMJ), foot contact time, and other related parameters were recorded using a force platform (9290AA; Kistler, Winterthur, Switzerland). To perform the CMJ, the participants were asked to stand on the force platform and place their hands on their hips. The subject then performed a rapidly downward squat movement and jumped vertically to attain maximum height. Arm-swing was not allowed during the jump.

Three trials separated by 1 min of passive recovery were performed. The best trial for jump height was included in the data analysis.

Reactive strength, normally by measured by DJ height and reactive strength index (RSI) (*Suchomel, Nimphius & Stone, 2016*), is defined as a runners' capacity to efficiently utilize the SSC and elastic energy produced by the muscle–tendon complex (*Beattie et al., 2017*). For the DJ test, all participants were asked to stand on a 40 cm-high box and place their hands on their hips. The participants then stepped off the box to land on the force plate and jumped vertically for maximum height and minimum ground contact time. The trial was successful only when the participants did not bend the hip or knee during the jump and their hands did not leave the hips. Three trials separated by 1 min of passive recovery were performed. The best trial for jump height was included in the data analysis. The RSI was calculated by dividing jumping height in cm by contact time in seconds.

### Running economy and related physiological tests

All physiological variables ($VO_{2max}$, RE, and blood lactate concentration, [BLa]) were measured using the treadmill protocol (Life Fitness T5, Rosemont, Illinois, USA). Oxygen uptake and heart rate (HR) were determined using a portable metabolic analyzer (K5, Cosmed Srl, Rome, Italy) and HR monitor belt (Garmin, Olathe, Kansas, USA). Finger blood, was collected prior to the test, was used to determine if the participant was in a normal state. The subject then warmed-up on the treadmill set to 8 $km.h^{-1}$ for 10 mins. After the warm-up period, the subject rested for 5 mins and then began a 4-minute run at each of 3 incremental speeds (12, 14, and 16 $km.h^{-1}$, respectively) to determine RE, which was defined as the average $VO_2$ ($ml.kg^{-1}.min^{-1}$) data during the last minute of each running speed. This RE testing protocol was similar to that in previous studies (*Cole et al., 2006*; *Sedano et al., 2013*) and reflected the runners' ability to run at submaximal speeds. Participants ran for 4 min to ensure adequate time for their $VO_2$, HR, and BLa to reach a steady-state (*Beattie et al., 2017*; *Saunders et al., 2006*). After each 4-minute stage of running, the subject rested for 1 min, during which finger blood samples were collected. All finger blood samples were used to measure their blood lactate concentrations and aerobic capacity during running via a lactate analyzer (EKF Diagnostic, Magdeburg, Germany).

After the completion of the last stage of the running economy test, the treadmill speed was set to 17 $km.h^{-1}$, which was then increased by 1 $km.h^{-1}$ every 2 mins, until the subject reached exhaustion. The following criteria were used to determine exhaustion: heart rate greater than 90% of age-predicted maximal HR (calculated by 220 –runner's age); respiratory exchange ratio (RER) $\geq 1.10$; and rating of perceived exertion (RPE) above 18. $VO_{2max}$ was determined as the highest $VO_2$ value using a 30s moving window.

### 50-m sprint and 5-km running performance test

For testing the runner's maximal speed and endurance performance, the 50-m and 5-km running tests were performed on a 200-m indoor track with an ambient temperature of 20–22 °C and a relative humidity of 61–64%. Following a 15-min warm-up, the participants performed three 50-m maximal effort sprint trials with a standing start position. The sprint time was measured via two timing gates (Smart speed, Fusion Sport, Brisbane, Australia) and the best trial for sprint time was included in the data analysis. After a 10-mins rest, for

**Table 2  Training arrangement during 8-week intervention.**

|  | Monday | Tuesday | Wednesday | Thursday | Friday | Saturday | Sunday |
|---|---|---|---|---|---|---|---|
| Moring | Road running: 15–20 km (75–85%HR$_{MAX}$) | IT:1,000 m*5sets (90–95%HR$_{MAX}$), work rest ratio 1:1 | Road running: 15–20 km (75–85%HR$_{MAX}$) | IT:1,000 m*5sets (90–95%HR$_{MAX}$), work rest ratio 1:1 | Road running: 15–20 km (75–85%HR$_{MAX}$) | IT:1,000 m*5sets (90–95%HR$_{MAX}$), work rest ratio 1:1; Road running: 10 km (70–80%HR$_{MAX}$) | Rest |
| Afternoon | Strength training |  | Strength training |  | Strength training |  |  |

Notes.

HR$_{MAX}$, Maximal heart rate;  IT,  Interval training.

ease and accuracy when measuring the trial time, the participants were randomly divided into four groups of seven participants, and instructed to finish the 5-km running test as fast as possible. Verbal encouragement was provided throughout the test. The trial time was recorded using a stopwatch (Tianfu, Shenzhen, Guangdong, China).

## Training intervention

During the training intervention, the participants were asked to perform 9 training sessions per week (six endurance and three strength sessions) with the schedule provided in Table 2. Participants performed different strength training programs according to their group allocation (Table 3). Participants were asked to avoided any other strength training besides the training intervention. All strength sessions were supervised by certified strength and conditioning specialists.

### Complex training group

This group performed three sets of three complex pairs (the pairing of two biomechanically similar exercises), including a back squat + a DJ from a 40 cm box (pair 1); a Bulgarian squat + a single leg hop (pair 2); and a Romanian deadlift + a double leg 50 cm hurdle hop (pair 3). All of the strength training performed in the complex pairs involved heavy loads (80–85%1RM) based on the pre-1RM strength test to enhance the power output of the subsequent plyometric exercises (*Carter & Jeremy, 2014*). The intra-complex rest interval (ICRI) was set at 4 mins since this is the optimal time for exploiting power according to previous studies (*Comyns et al., 2006*).

### Heavy resistance training group

This group performed the identical lower limb strength training as the CT group, but excluded the plyometric exercises. In order to ensure that the CT and HRT workloads were similar, the HRT group added two additional sets for each strength exercises (same intensity and repetition) to match the plyometric training load in the CT group. The recovery time between sets was 3 mins as this is the optimal time for improving maximum strength in distance runners (*Storen et al., 2008*).

**Table 3  Strength training program during study.**

| Exercise | Load | Sets | Repetition | Recovery (mins) |
| --- | --- | --- | --- | --- |
| *Preparatory strength training* | | | | |
| Back squat | 20 kg | 3 | 10 | 3 |
| Bulgarian squat | 20 kg | 3 | 5 ES | 3 |
| Romanian deadlifts | 20 kg | 3 | 10 | 3 |
| *Training intervention* | | | | |
| **CT group** | | | | |
| Back squat | 80–85%1RM | 3 | 5 | 4 |
| Drop jump | 40 cm Box | 3 | 6 | 4 |
| Bulgarian squat | 80–85%1RM | 3 | 5 ES | 4 |
| Single leg hop | BW | 3 | 6 ES | 4 |
| Romanian deadlifts | 80–85%1RM | 3 | 5 | 4 |
| Double leg hurdle hop | 50 cm Hurdle | 3 | 6 | 4 |
| **HRT group** | | | | |
| Back squat | 80–85%1RM | 5 | 5 | 3 |
| Bulgarian squat | 80–85%1RM | 5 | 5 ES | 3 |
| Romanian deadlift | 80–85%1RM | 5 | 5 | 3 |
| **CON group** | | | | |
| Back squat | 40%1RM | 5 | 20 | 1 |
| Bulgarian squat | 40%1RM | 5 | 15 ES | 1 |
| Romanian deadlift | 40%1RM | 5 | 20 | 1 |

**Notes.**
RM, maximal repetition; BW, body weight; ES, each side.

### Control group

This group performed the strength-endurance regimen. The resistance exercises the CON group used were as same as the CT and HRT groups, but the load was reduced to 40% 1RM and the repetitions were increased to 20–30. In addition, the CON group added two more sets for each strength exercises (same intensity and repetition) to match the plyometric training or heavy resistance training load. The recovery time between sets was 1 min because a muscular strength-endurance training program has very short rest period (*Sedano et al., 2013*).

### Endurance training

In addition to strength training, all three groups performed the same endurance training, which mainly consisted of long-distance road running at an intensity of 70–85% maximal heart rate ($HR_{MAX}$) and interval training at an intensity of 90–95% $HR_{MAX}$. The total endurance training distance was $77.25 \pm 2.33$ km per week, and the total endurance and strength training times were $8.75 \pm 0.97$ and $3.5 \pm 0.5$ h /week, respectively.

## Statistical analyses

Statistical analyses were performed using SPSS 22.0 (IBM Corp. Armonk, NY, USA). Means $\pm$ standard deviations (SD) are shown in Table 4–6. Normality and homogeneity of variances were tested via Shapiro–Wilk and Levene's tests, respectively. All variables were

**Table 4  Results of body composition before and after intervention (Mean ± SD).**

|  | Complex training group (N = 10) | | | Heavy resistance group (N = 9) | | | Control group (N = 9) | | |
|---|---|---|---|---|---|---|---|---|---|
|  | Before | After | %Δ | Before | After | %Δ | Before | After | %Δ |
| BM (kg) | 63.08 ± 6.08 | 63.44 ± 6.85 | 0.53 ± 3.71 | 57.76 ± 3.70 | 57.92 ± 3.05 | 0.39 ± 2.64 | 61.36 ± 4.49 | 61.27 ± 4.69 | −0.16 ± 0.97 |
| BMI (kg.m$^{-2}$) | 19.86 ± 1.61 | 19.84 ± 1.66 | −0.03 ± 4.33 | 18.77 ± 1.03 | 18.86 ± 0.86 | 0.43 ± 2.46 | 19.24 ± 0.81 | 19.27 ± 0.95 | 0.10 ± 1.54 |
| FFM (kg) | 55.99 ± 4.86 | 56.39 ± 5.39 | 0.66 ± 1.86 | 52.26 ± 3.04 | 52.10 ± 2.85 | −0.24 ± 2.90 | 55.97 ± 4.02 | 55.74 ± 4.10 | −0.40 ± 0.82 |
| FM (kg) | 7.09 ± 3.14 | 7.05 ± 3.16 | 3.88 ± 6.29 | 5.50 ± 1.99 | 5.82 ± 2.13 | 6.69 ± 13.52 | 5.39 ± 1.70 | 5.52 ± 1.95 | 1.60 ± 7.92 |

Notes.
BM, body mass; BMI, body mass index; FFM, fat-free mass; FM, fat mass.

**Table 5  Results of strength and power assessment before and after intervention (Mean ± SD).**

|  | Complex training group (N = 10) | | | Heavy resistance group (N = 9) | | | Control group (N = 9) | | |
|---|---|---|---|---|---|---|---|---|---|
|  | Before | After | %Δ | Before | After | %Δ | Before | After | %Δ |
| 1RM (kg) | 60.25 ± 8.03 | 70.50 ± 11.17[***] | 16.88 ± 5.93 | 60.56 ± 11.84 | 71.67 ± 12.50[***] | 18.80 ± 6.42 | 63.33 ± 9.35 | 64.44 ± 8.82 | 2.15 ± 6.64 |
| CMJ Height (cm) | 31.06 ± 3.41 | 34.51 ± 3.85[***] | 11.28 ± 7.57 | 32.8 ± 4.23 | 35.58 ± 3.33[***] | 8.96 ± 4.94 | 33.46 ± 4.27 | 34.26 ± 4.22 | 2.46 ± 1.64 |
| Peak Power (w.kg$^{-1}$) | 43.66 ± 2.70 | 47.12 ± 2.65 | 8.20 ± 7.79 | 45.18 ± 3.57 | 47.45 ± 5.33 | 5.05 ± 8.69 | 45.08 ± 3.17 | 45.29 ± 3.01 | 0.53 ± 3.66 |
| DJ Height (cm) | 31.39 ± 4.41 | 35.38 ± 4.55[***] | 12.94 ± 4.67 | 33.03 ± 3.79 | 33.96 ± 4.13 | 2.88 ± 6.58 | 33.31 ± 4.53 | 32.81 ± 4.26 | −1.40 ± 1.74 |
| RSI (cm.s$^{-1}$) | 59.05 ± 11.63 | 70.80 ± 15.69[***] | 19.99 ± 12.39 | 61.57 ± 12.95 | 66.71 ± 8.53 | 10.43 ± 15.69 | 62.11 ± 12.04 | 62.91 ± 12.07 | 1.41 ± 5.70 |

Notes.
1RM, one repetition maximum back squat; CMJ, counter movement jump height; DJ, drop jump height; Peak Power, peak power in counter movement jump; RSI, reactive strength index.
*$p < 0.0167$, **$p < 0.01$, ***$p < 0.001$ significant difference from pre- to post-test.

tested using a one-way analysis of variance (ANOVA) to determine differences among the three training groups. Two-way repeat measures were performed using ANOVA with time (pre- vs. post-test) and group (CT vs. HRT vs. CON) as factors to assess training-related effects. When a statistically significant difference was found for time by group interaction effects or main effects ($p \leq 0.05$), pairwise comparisons were performed using a post hoc $t$ test with Bonferroni correction. For effect size, partial eta-squared was calculated ($\eta 2$), and 0.2, 0.5, and 0.8 were interpreted as small, medium and large effect sizes, respectively (*Cohen, 1988*). An alpha level of $p < 0.0167$ was set for establishing statistical significance.

# RESULTS

All pre- and post-test variables are shown in Table 4–6. No differences were found in any of these variables among the CT, HRT, and CON groups during the pre-test. Body mass, BMI, fat-free mass, fat mass and fat mass percentages remained unaltered in all three groups from pre- to post-test (Table 4).

Li et al. (2019), *PeerJ*, DOI 10.7717/peerj.6787

Peer**J**

**Table 6  Results of physiological and running performance assessment before and after intervention (Mean ± SD).**

| | Complex training group (N = 10) | | | Heavy resistance group (N = 9) | | | Control group (N = 9) | | |
|---|---|---|---|---|---|---|---|---|---|
| | **Before** | **After** | **%Δ** | **Before** | **After** | **%Δ** | **Before** | **After** | **%Δ** |
| $VO_{2max}$ (ml.kg$^{-1}$.min$^{-1}$) | 65.65 ± 5.06 | 64.47 ± 4.31 | −1.57 ± 5.78 | 65.54 ± 5.52 | 64.65 ± 6.18 | −1.25 ± 6.58 | 66.14 ± 5.25 | 67.79 ± 3.03 | 2.81 ± 5.05 |
| RE at 12 km.h$^{-1}$ (ml.kg$^{-1}$.min$^{-1}$) | 46.62 ± 3.20 | 44.42 ± 2.32[**] | −4.47 ± 5.92 | 47.01 ± 3.14 | 45.06 ± 3.82[*] | −4.11 ± 5.33 | 47.81 ± 3.24 | 45.71 ± 3.05[*] | −4.38 ± 1.59 |
| RE at 14 km.h$^{-1}$ (ml.kg$^{-1}$.min$^{-1}$) | 54.45 ± 3.18 | 50.17 ± 3.04[***] | −7.68 ± 6.33 | 53.41 ± 3.26 | 50.78 ± 4.07[**] | −4.89 ± 5.26 | 53.95 ± 2.27 | 52.33 ± 2.31 | −2.98 ± 2.62 |
| RE at 16 km.h$^{-1}$ (ml.kg$^{-1}$.min$^{-1}$) | 60.77 ± 4.17 | 56.09 ± 2.55[***] | −7.38 ± 6.67 | 59.60 ± 3.04 | 57.88 ± 4.11 | −2.91 ± 4.21 | 61.00 ± 3.00 | 61.35 ± 1.79 | 0.72 ± 3.99 |
| BLa at 12 km.h$^{-1}$ (mmol.L$^{-1}$) | 1.46 ± 0.43 | 1.29 ± 0.31 | −1.93 ± 43.93 | 1.44 ± 0.46 | 1.24 ± 0.40 | −12.72 ± 16.13 | 1.43 ± 0.53 | 1.37 ± 0.39 | 2.60 ± 30.62 |
| BLa at 14 km.h$^{-1}$ (mmol.L$^{-1}$) | 1.60 ± 0.63 | 1.29 ± 0.43 | −12.50 ± 30.80 | 1.62 ± 0.73 | 1.48 ± 0.53 | 1.12 ± 37.82 | 1.67 ± 0.35 | 1.32 ± 0.36 | 45.06 ± 56.99 |
| BLa at 16 km.h$^{-1}$ (mmol.L$^{-1}$) | 2.56 ± 0.48 | 1.45 ± 0.45[***] | −40.80 ± 14.82 | 2.30 ± 0.53 | 1.94 ± 0.68 | −16.05 ± 17.89 | 2.76 ± 0.59 | 2.41 ± 0.51 | −8.09 ± 32.15 |
| HR at 12 km.h$^{-1}$ (beats.min$^{-1}$) | 136.34 ± 7.97 | 136.14 ± 8.80 | −0.07 ± 5.00 | 138.86 ± 12.92 | 136.54 ± 8.74 | −1.29 ± 6.19 | 142.07 ± 13.59 | 143.54 ± 15.12 | 1.10 ± 6.21 |
| HR at 14 km.h$^{-1}$ (beats.min$^{-1}$) | 149.08 ± 8.60 | 149.28 ± 9.32 | 0.18 ± 4.13 | 153.13 ± 14.41 | 149.16 ± 9.84 | −2.29 ± 4.86 | 155.34 ± 13.22 | 156.03 ± 14.83 | 0.53 ± 6.34 |
| HR at 16 km.h$^{-1}$ (beats.min$^{-1}$) | 164.81 ± 8.31 | 160.74 ± 7.51 | −2.43 ± 1.83 | 166.95 ± 13.77 | 164.41 ± 10.39 | −1.35 ± 3.71 | 169.59 ± 14.78 | 169.93 ± 16.33 | 0.31 ± 6.72 |
| 5-km time (s) | 953.70 ± 12.30 | 926.90 ± 9.92[*** ##] | −2.80 ± 0.87 | 952.56 ± 10.10 | 932.67 ± 11.61[***] | −2.09 ± 0.52 | 954.11 ± 6.75 | 947.33 ± 10.03 | −0.07 ± 1.51 |
| 50-m time (s) | 6.25 ± 0.19 | 6.11 ± 0.24[**] | −2.26 ± 2.16 | 6.17 ± 0.33 | 6.04 ± 0.33[**] | −2.14 ± 2.35 | 5.94 ± 0.21 | 5.92 ± 0.30 | −0.4 ± 2.03 |

**Notes.**

BLa, blood lactate concentrations;  HR, heart rate;  RE, running economy;  $VO_{2max}$, maximum oxygen uptake.

[*]p<0.0167.

[**]p<0.01.

[***]p<0.001 significant difference from pre- to post-test.

[##]p<0.01, significantly different from control group at post-test.

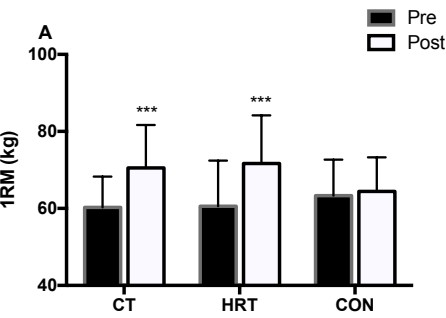
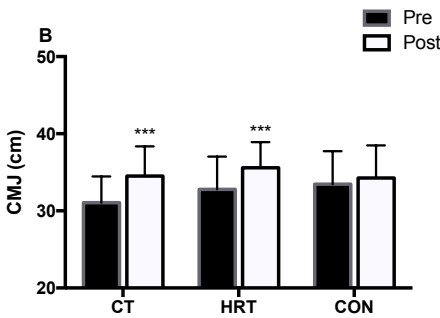

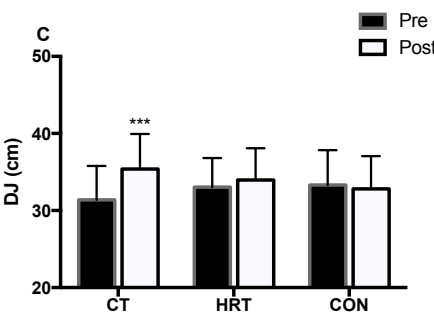
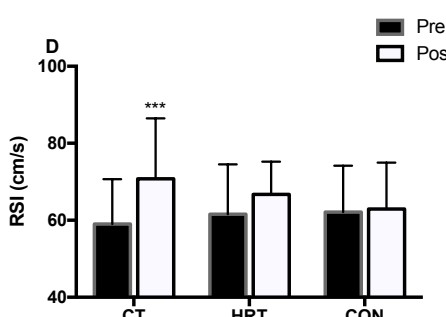

**Figure 2** **Neuromuscular adaptation over 8 weeks of intervention.** 1RM (A), CMJ (B), DJ (C) and RSI (D) over 8 weeks of intervention (Mean ± SD). 1RM, one repetition maximum strength; CMJ, counter movement jump height; DJ, drop jump height; RSI, reactive strength index. *** $p < 0.001$ significant difference from pre- to post-test.

## Strength and power test

The two-way repeated measures ANOVA showed a significant time by group interaction or time main effect on 1RM ($F_{(2,25)} = 17.414$, $p < 0.001$, $\eta2 = 0.578$), CMJ height ($F_{(2,25)} = 7.404$, $p = 0.03$, $\eta2 = 0.372$), peak power ($F_{(1,25)}11.589$, $p = 0.002$, $\eta2 = 0.317$), DJ height ($F_{(2,25)} = 25.355$, $p < 0.001$, $\eta2 = 0.670$), and RSI ($F_{(2,25)} = 4.914$, $p = 0.016$, $\eta2 = 0.282$). The Bonferroni post hoc test showed that there were significant increases between the pre- and post-tests in: 1RM strength (CT: 16.88%, $p < 0.001$; HRT: 18.80%, $p < 0.001$) and CMJ height (CT: 11.28%, $p < 0.001$; HRT:8.96%, $p < 0.001$), but not for peak power (CT: 8.20%, $p = 0.02$; HRT: 5.05%, $p = 0.036$) in the CT and HRT groups. Only the CT group showed a significant increase between the pre- and post-test for DJ height (12.94%, $p < 0.001$) and RSI (19.99%, $p < 0.001$). No significant changes were observed in the CON group for these indicators (Fig. 2).

## Running economy and related physiological test

Table 6 demonstrated that there were no significant differences in $VO_{2max}$ in the time by group interaction or main effects after the 8-week intervention. For the RE test, there was a

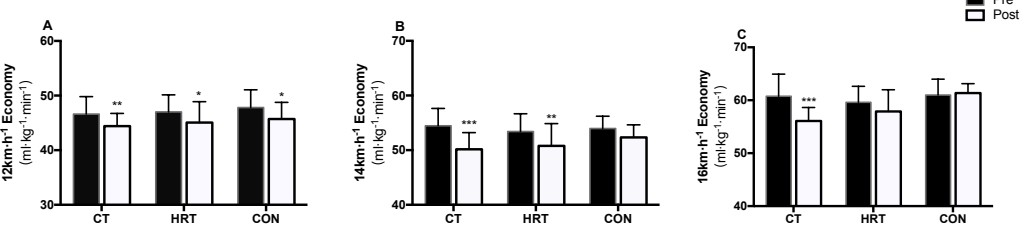

**Figure 3** **Running economy changes over 8 weeks of intervention.** Running economy at (A) 12, (B) 14 and (C)16 km.h$^{-1}$ over 8 weeks of intervention (Mean $\pm$ SD). *$p < 0.0167$, **$p < 0.01$, **$p < 0.001$ significant difference from pre- to post-test.

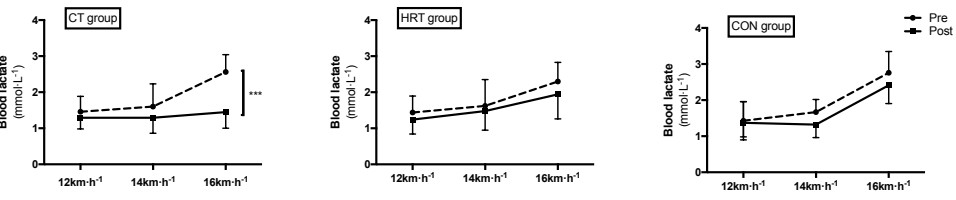

**Figure 4** **Blood lactate concentrations changes over 8 weeks of intervention.** Blood lactate concentrations at the speeds of 12, 14, 16 km.h$^{-1}$ over 8 weeks of intervention (Mean $\pm$ SD). ***$p < 0.001$, significant difference from pre- to post-test.

significant interaction between time and group on 16 km.h$^{-1}$RE ($F_{(2,25)} = 5.646$, $p = 0.009$, $\eta2 = 0.311$). The Bonferroni post hoc test found that there was a significant improvement between the pre- and post-test only in the CT group ($-7.38\%$, $p < 0.001$). There was a significant time main effects for 12 km.h$^{-1}$ RE ($F_{(1,25)} = 23.554$, $p < 0.001$, $\eta2 = 0.485$) and each group had a significant improvement between the pre- and post-test (CT: $-4.47\%$, $p = 0.005$; HRT: $-4.11\%$, $p = 0.016$; CON: $-4.38\%$, $p = 0.01$). For the 14 km.h$^{-1}$ RE, a significant time main effect ($F_{(1,25)} = 28.9244$, $p < 0.001$, $\eta2 = 0.536$) was observed: the CT ($-7.68\%$, $p < 0.001$) and HRT groups ($-4.89\%$, $p = 0.009$) showed a significant improvement between the pre- and post-test (Fig. 3). In addition, there was a significant time by group interaction in BLa at 16 km.h$^{-1}$ ($F_{(2,25)} = 4.2182$, $p = 0.026$, $\eta2 = 0.252$). The Bonferroni post hoc test identified that the BLa at 16 km.h$^{-1}$ was significantly decreased in the CT group ($-40.80\%$, $p < 0.001$) (Fig. 4). There were no significant time by group interactions or main effects on HR at any speed after the intervention.

## 5-km running and 50-m sprint performance test

There was a statistically significant time by group interaction in 5-km running performance ($F_{(2,25)} = 9.627$, $p = 0.001$, $\eta2 = 0.435$). The Bonferroni post hoc test showed that there were significant improvements between the pre- and post-tests in the CT ($-2.80\%$, $p < 0.001$) and HRT ($-2.09\%$, $p < 0.001$) groups, and the CT group was significantly lower when compared to the CON group at the post-test ($p = 0.001$). There was a significant time main effects for the 50-m sprint performance ($F_{(1,25)} = 14.903$, $p = 0.001$, $\eta2 = 0.373$), and the CT ($-2.26\%$, $p = 0.003$) and HRT groups ($-2.14\%$, $p = 0.007$) showed a significant

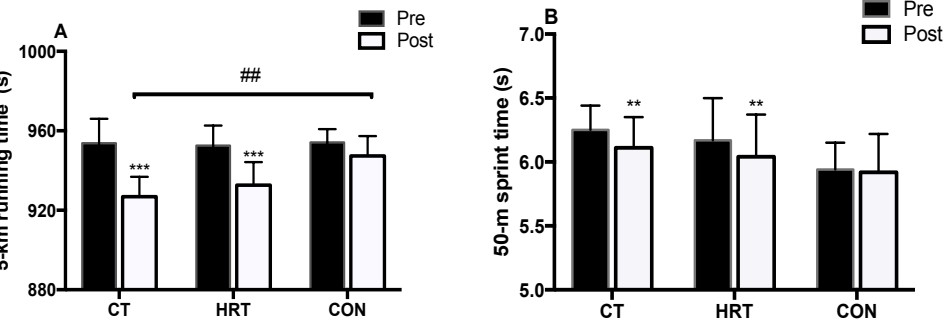

**Figure 5  5-km running and 50-m sprint performance changes over 8 weeks of intervention.** 5-km running (A) and 50-m sprint (B) performance over 8 weeks of intervention (Mean ± SD). **$p < 0.01$, ***$p < 0.001$, significant difference from pre- to post-test; ##$p < 0.01$, significant difference from CON group.

improvement between the pre- and post-tests (Fig. 5). There were no significant changes in the CON group in terms of 5-km running or 50-m sprint performance after the intervention.

## DISCUSSION

The aim of this study was to determine the effects of CT vs. HRT on neuromuscular adaption, RE and 5-km performance in well-trained male distance runners. We hypothesized that CT would elicit greater training benefits than HRT, which was supported by the results of this study. The primary finding of this study was that performing CT or HRT with endurance training significantly increased strength and power performance, RE at 12 and 14 km.h$^{-1}$, 50-m sprint times, and 5-km performance in well-trained male distance runners. Furthermore, CT demonstrated a greater benefit in terms of reactive strength, 16 km.h$^{-1}$ RE and reduction of BLa concentrations at 16 km.h$^{-1}$. Long-distance runners could integrate this time-efficient training approach into their training program for improving running performance.

### Neuromuscular adaptation

Neuromuscular adaptations, including 1 RM strength and jumping ability, were especially important indicators of increasing athletic capacity in strength and power athletes, and recent studies showed that they also have a positive impact on modifying distance running performance such as increasing force production during running and increasing RE (*Berryman et al., 2017*; *Stone, Stone & Sands, 2006*). After the training intervention, there was a significant increase in 1RM strength for participants in the CT (16.88%) and HRT groups (18.80%), but not in the CON group. In accordance with our results, previous studies have demonstrated similar improvements (12–33.2%) in 1 RM strength after 8 weeks of HRT among distance runners (*Damasceno et al., 2015*; *Storen et al., 2008*). These results were consistent because the similar heavy load strength programs (i.e., 3–5 RM) were adopted in previous studies. Heavy strength training exerts high mechanical stimulus on the neuromuscular system, thus producing good effects on maximal strength

development. It has been reported that high load strength training (3–6 RM) is associated with maximal force generation without muscle hypertrophy, and is attributed to neural adaptation (*Docherty & Sporer, 2000*). This is consistent with our results. Considering that no changes in body weight, fat-free mass, and BMI were observed, we hypothesized that the improved maximum strength likely resulted from neural adaptation such as motor unit recruitment and increase in firing frequency (*Cormie, Mcguigan & Newton, 2011*). This was a concern for distance runners because increased body mass negatively impacts physiological parameters, such as $VO_{2max}$ or RE, and inevitably impairs endurance development (*Beattie et al., 2017*).

With respect to power performance, the CT and HRT groups showed a significant increase in CMJ height (11.28% and 8.96%, respectively) after the intervention. The CMJ height increase in the CT group was higher than the improvements (4.5–8.9%) noted in previous studies which used HRT or PLY training interventions in distance runners with similar intervention timeframes (*Ramírezcampillo et al., 2014*; *Vikmoen et al., 2016*). This may be attributed to the different strength training protocols in the training arrangement. Previous research has confirmed the positive effects of CT on increasing power performance and may be more efficient than using one strength training method (*Macdonald, Lamont & Garner, 2012*). The mechanism underpinning this phenomenon may be attributable to HRT causing increases in motor neuron excitability, thus creating optimal conditions for subsequent plyometric exercises (*Esformes & Bampouras, 2013*). The cumulative effect of CT would then lead to greater power output (*Jones & Lees, 2003*).

Reactive strength is defined as the ability of the muscle–tendon complex to produce greater force during the concentric contraction immediately following a rapid eccentric contraction (*Beattie et al., 2016*). These indicators, such as DJ and RSI, are very important for distance running performance (*Beattie et al., 2017*). In the present study, the CT group showed a significant improvement in DJ performance (12.94%) and RSI (19.99%) between the pre- and post-tests, whereas no changes were noted in the HRT and CON groups. The advantageous CT effect on reactive strength can be explained by the inclusion of plyometric exercises such as drop jump, single leg hops, and hurdle hops in CT. These exercises exerted maximal external force on the neuromuscular system in a very short time, greatly enhancing power by developing SSC utilization and leg musculotendinous stiffness (*Saunders et al., 2006*; *Spurrs, Murphy & Watsford, 2003*). Interestingly, our results are also consistent with those of the study by *Ramírezcampillo et al., (2014)*, who reported improvement in DJ performance by 16.7% after 6 weeks of plyometric training in competitive distance runners. Although the plyometric exercise volume in both this study and ours was relatively small (60 vs.72 contacts per session), the training appears sufficient in distance runners, who may not be accustomed to such exercises. Recent research supported that a low-volume plyometric program can produce similar performance improvements in terms of reactive strength compared with that in a high-volume plyometric program (*Jeffreys et al., 2017*). Taken together, our results support the hypothesis that CT increases neuromuscular facilitation and is appropriate for long-distance runners and may be more efficient than HRT and PLY training methods.

## Running economy

Running economy is described as the steady-state oxygen consumption at a given running speed and is a critical physiological measurement to determine distance running success (*Saunders et al., 2004*). After training intervention, participants in the CT and HRT groups showed a significant improvement in 12 $km.h^{-1}$ RE (4.47 vs. 4.11%) and 14 $km.h^{-1}$ RE (7.68 vs. 4.89%). Our findings are consistent with those of previous research that reported similar increases in RE after heavy strength interventions for the same level distance runners (*Storen et al., 2008*). The RE improvement in the CT and HRT groups could be attributed to neuromuscular adaptations after heavy strength training. The multi-joint and closed-chain high load exercises, such as back squat, increases maximum force and peak power, theoretically allowing the athletes to maintain constant speed or to perform each running action at a relatively lower force ratio (*Aagaard & Andersen, 2010*; *Ronnestad & Mujika, 2014*). Meanwhile, strength training may potentiate the neural recruitment and function of type I muscle fibers as well as postponing the activation of the less efficient type II fibers (*Schumann et al., 2016*), therefore reducing overall energy consumption. In terms of changes in 16 $km.h^{-1}$ RE, only the CT group showed a significant improvement (−7.38%) after training intervention. This may indicate that integrating CT or HRT training into endurance training programs can elicit similar improvements of RE at low speed; as speed increases, the role of CT becomes more substantial. In line with our results, Sedano et al. (*Sedano et al., 2013*) combined three jumping exercises into a strength training program (70%1RM) and showed significantly improvements in well-trained runners' RE at 16 $km.h^{-1}$. Therefore, it can be hypothesized that reactive strength adaptation plays an important role in RE improvement at higher speeds. The elastic energy stored during the eccentric contractions of running and recoiled following concentric contractions makes an extensive contribution to propulsion (*Anderson, 1996*), and may even exceed the efficiency of conversion from chemical energy to kinetic energy by the muscle (*Williams, 1985*). The complex pair exercise in CT may enhance the runners' reactive strength and the ability to store and utilize elastic energy by facilitating the SSC, it theoretically improving the leg stiffness and reducing the time the athletes' foot spends in contact with ground, thereby decreasing the energy cost during running. As running speed increases, elastic mechanisms prevail over the muscle contractile capabilities and may even account for a bigger proportion of the total work (*Saunders et al., 2006*).

In addition, the CT group showed a decrease in BLa concentrations at 16 $km.h^{-1}$, which means that athletes could run at the same speed with smaller anaerobic energy consumption. The potential explanation may be that CT combined with endurance training increases the strength and function of type I muscle fibers while delaying the recruitment of type II muscle fibers (*Marcinik et al., 1991*; *Midgley, McNaughton & Jones, 2007*); thus, reducing lactate production and improving the capability for dispelling lactate at higher running velocities (*Jones & Carter, 2000*). Therefore, CT can achieve the HRT and PLY training effects at the same time leading to significantly increased RE and decreased BLa concentrations at high speeds.

## Running performance

The goal of athletic training is to create meaningful improvements in performance. This study revealed that both the CT and HRT groups showed a significant improvement (−2.8% vs. −2.09%) in 5-km running performance, and the CT group was significantly lower than that in the CON group. The magnitude of the improvement was in line with that in previous studies, which have shown that the increase in running performance was accompanied by augmentation of neuromuscular adaptations and running economy (*Berryman et al., 2017*; *Damasceno et al., 2015*; *Ramírezcampillo et al., 2014*). For example, Ramírezcampillo et al. (*Ramírezcampillo et al., 2014*), demonstrated that jumping and 2.4 km running performance in highly-trained athletes improved simultaneously after performing 8 weeks of plyometric and endurance training. *Damasceno et al. (2015)*, found that the addition of 8 weeks of heavy strength training to endurance programs resulted in a significant enhancement in 10-km time trials. Since there was no significant change for $VO_{2max}$, we speculated that the 5-km performance improvements in the HRT and CT groups were due to the comprehensive effects of neuromuscular adaptations, RE increase, and the decrease in BLa at constant speeds. In addition, strength and plyometric exercises (back squat, Bulgarian squat and single leg hops, etc.) in the CT group, which involved similar kinematic traits to a running gait, likely provided the greatest improvement in endurance performance (*Bazyler et al., 2015*). On the other hand, the participants in the CT and HRT groups achieved a significant improvement in their 50-m sprint performance (−2.26% and −2.14% respectively) whereas no significant improvements were found in the CON group. This is in line with that in previous studies, which showed improvements of 1.1–3.4% following various strength training interventions in distance runners (*Mikkola et al., 2007*; *Paavolainen, Nummela & Rusko, 1999*; *Ramírezcampillo et al., 2014*). The enhancement of sprint performance plays an important role in distance races and allows runners to hold a favorable position at the start and to accelerate at the end of the race. These findings highlighted the necessity of utilizing a multicomponent conditioning program, such as CT, for developing endurance performance.

It is worth noting that the present study only enrolled young male distance runners; therefore, the results may not necessarily be broadly applicable to all types of runners. Since CT was at a relatively high intensity, coaches should be prudent when using this type of training. The age, gender and strength level must be considered before the onset of training. In addition, as HRT has similar training effects on RE and neuromuscular adaptations, coaches could use this type of training to replace the CT when the training subjects are female or older runners. We suggest that a training block of CT could follow a block of HRT so that basic strength levels could be developed. In future studies, we need to investigate the effects of CT on different types of runners (e.g., female or older runners) for longer periods of time (>16 weeks) to optimize endurance performance.

## CONCLUSIONS

In conclusion, we found that compared with strength-endurance training, combining 8 weeks of CT or HRT with endurance training resulted in enhanced effects on 1RM strength,

CMJ height, running economy, maximal sprint speed and 5-km running performance in well-trained male distance runners. In addition, CT is an efficient method to combine the advantages of HRT and PLY exercises at the same time, exhibiting greater improvements in terms of reactive strength, 16 km.h$^{-1}$RE, and reduction of BLa concentrations at a speed of 16 km.h$^{-1}$. Long-distance runners could integrate CT into their programs to improve the running performance. For well-trained male runners who have limited time to do strength training, three complex-pair exercises three times a week with a relatively small PLY volume, could optimize their endurance performance and is superior to HRT alone.

## ACKNOWLEDGEMENTS

The authors would like to thank all the runners who were involved in this research and all the members and researchers at the Chinese Marathon College, Chinese Athletic Association and Strength and Conditioning Research Center of Shanghai University of Sport for their contribution to this study.

### Funding

This study was funded by the scientific and technological research program of the Shanghai Science and Technology Committee (Fund number:17080503300). The funders had no role in study design, data collection and analysis, decision to publish, or preparation of the manuscript.

### Grant Disclosures

The following grant information was disclosed by the authors:
Scientific and technological research program of Shanghai Science and Technology Committee: 17080503300.

### Competing Interests

The authors declare there are no competing interests.

### Author Contributions

- Fei Li conceived and designed the experiments, performed the experiments, analyzed the data, contributed reagents/materials/analysis tools, prepared figures and/or tables, authored or reviewed drafts of the paper, approved the final draft.
- Ran Wang conceived and designed the experiments, authored or reviewed drafts of the paper.
- Robert U. Newton and David Sutton analyzed the data, authored or reviewed drafts of the paper.
- Yue Shi performed the experiments, prepared figures and/or tables.
- Haiyong Ding conceived and designed the experiments, approved the final draft.

## Human Ethics

The following information was supplied relating to ethical approvals (i.e., approving body and any reference numbers):

Ethics Committee of Shanghai University of Sport granted ethical approval to carry out the study within its facilities (Approval form No. 2017047)

## Data Availability

The raw data are available in the Supplemental Materials.

## Supplemental Information

Supplemental information for this article can be found online at http://dx.doi.org/10.7717/peerj.6787#supplemental-information.

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
