# Peer review of "Effects of complex training versus heavy resistance training on neuromuscular adaptation, running economy and 5-km performance in well-trained distance runners"

_PeerJ, doi:10.7717/peerj.6787_

## Round 0.1 · original submission · Major Revisions

The reviewers and I agree that the manuscript contains interesting experimental data on training relative to endurance running performance. The majority of problems identified are mainly in presentation so I believe a revision will not be too time consuming—unless the issue I raise about the control group needs to be dealt with (see below). I also caution the authors to pay particular attention to specific reviewer comments in addition to my summary comments here. Please note that one reviewer has uploaded a marked PDF with detailed comments in addition to the review. The writing and formatting (please spell out words and do not use acronyms) need editing but beyond that, presentation of data and clarification of study details also require attention.

The reviewers and I note that the results are not presented in a reader friendly format. This is particularly true starting with line 267 where we are confronted with several parenthetical test results. Please provide tables for each subhead when there are several tests to be discussed. One reviewer has strongly recommended graphical summary of key findings and I agree this would be very helpful for the reader to have visualized the various measures relative to training regimen.

There are some questions regarding the participants. Please state how they were recruited and clarify what you mean when you say there were all long-distance runners—are they professional athletes, hobbyists, part of a team? There are also some questions raised about the control group. You first state that controls engaged in their usual routines (line 123) but later you state that they were given the same exercises as the experimental group but at a reduced effort (line 237). The former approach is a typical control, particularly if the training regimen is shared by the larger group (assuming they are a track team) in normal conditions. If other approach mentioned is a problem because the data then can only be used to test the hypothesis that this training regimen works better at a greater effort, not that it works better compared to other types of training typically used. Please justify why they are engaging in a reduced effort version of the experimental group and alter your hypothesis testing and manuscript to reflect what the data are capable of demonstrating.s

A final major point is that the study is limited to males (and presumably male college athletes on a long-distance running team but this is not yet clear). As such, the findings cannot be generalized to members of the larger population let alone to non-males. Please revise the discussion, conclusion and abstract to reflect the limitations of the study.

If are you able to address these points and the specific comments of the reviewers, we are happy to consider a revised manuscript.

Reviewer 1 ·

Basic reporting

I believe that with some clarifications throughout the manuscript, that this work should be published.

However, language should be improved throughout the manuscript with particular attention to: subject-verb agreement, tense, and missing prepositions or articles. Examples include: lns 71, 75-76, 81-82, 148, 185, 205, 207-208, 238, 262, 281, 285, 301-302, 319, 320 (spelling), 323, 327 (don’t need “especially” there), 329, 335, 341, 355, 368 (word choice), 372, 376, 387, 397, 428 (word choice, “transfer”). This list of examples is not exhaustive.

Also, the authors inconsistently space references within the text. There should be a space between the last word and the citation.

Other specific notes:

Abstract: Overall, I think it would be beneficial to shorten the background portion and add some more detail to the methods and results. For example, it is not clear what your control group actually is from how it is currently written, and the reader is left wondering if the control group takes part in any training at all. Also, more detail regarding how you measured performance would be helpful. At the moment “performance” is a vague description. Please add detail on the specific performance tests – even if just the test names. For the results, do you mean there are difference relative to the control group? Or are there differences between the CXT and HRT groups? You need to clarify. It would also be helpful to clarify what kind of improvements (direction & magnitude) were made.

Key Words: I think it would be helpful to include a key word/phrase relevant to training programs

Introduction:
1. I think the first two paragraphs should be combined and shortened so you can get to your main point faster.
2. Line 87: please define RM (1 rep max) for the reader
3. Lines 92-93: How so? You need to make an argument how explosive power is important for endurance running. You make this point in the discussion (lns 433-435) – but this argument needs to be made in the introduction to tell the reader why explosive power, as developed through HRT and plyometrics, is actually important for endurance athletes.
4. It would be helpful for the authors to lay out the neuromuscular adaptations of interest earlier on in the introduction and why they are the ones of interest.

Experimental design

Methods - specific notes
1. Who were these participants? Part of a university team? National team? Where were they located?
2. Part of “Procedures – Preparatory Strength Training” – it would be helpful to know what exercises were employed – even just by referring to one of your already constructed tables.
3. The authors need to make the argument for which tests they are using and why they are using those tests much earlier on in the methods and procedures. This is done to some degree in the discussion (lns 348-353), but the rationale for the different strength and running tests needs to be provided early in the methods.
4. This is a style point, but it would be helpful visually in table 3 to bold or underline the different training group names in the first column – it is tough to pick out.

Validity of the findings

Results
1. Throughout the results, instead of just saying significant interaction or significant difference, it would be helpful to provide the direction and magnitude of those differences to give the reader a better idea of what is happening
2. Ln 302-305 – what about the control group?
3. The results overall could be written more clearly to truly lay out the differences between the group. It is not always readily apparent if there are differences between CXT and HRT groups relative to the control. A short summary paragraph of where there were differences (as I state in #1 above) and the magnitude of those differences would be helpful.

Discussion:
1. I would recommend not using the word “increment”. This word implies a regular (consistent) change in value. Using “increased” or “improvement” would be a better word choice.
2. Lns 385-386 – instead of saying different – say how they are different
3. Lns 401-403 – wouldn’t this be a sign of greater adaptation in Type I fibers? Endurance and efficiency improved so that type II fibers are not used? And the argument has not necessarily been made of better lactate removal – it could be that Type I fibers are more active and efficient.
4. Ln 424 – clarity – is this for all groups?

Additional comments

The authors are to be commended on the experimental design and extensive testing the performed.

Reviewer 2 ·

Basic reporting

There are a few spots in the paper in which it is not clear what the authors hope to convey (please see my annotated pdf where I note such spots and where I suggest grammatical and typographical edits). In terms of basic reporting, the raw data are supplied, the paper has a traditional structure, the hypotheses are clearly laid out, and the results of analyses are shown in tables. However, there are no figures showing the data. Given the many tests that were performed, I suggest that the authors should make a few figures that graphically demonstrate what they consider to be their most important results.

Experimental design

The general basis for the study is sound, and the experimental design is clearly explained.

Validity of the findings

There are a few areas in the Discussion section of the manuscript (please see my annotated pdf for specifics) where explanations for the study’s findings are hypothetical/speculative, but are not clearly stated as such. Please revise to specifically state which interpretations are speculative and which are supported by the data in this paper. Also, please revise the discussion to note that this study only examines young male endurance athletes and is therefore not necessarily broadly applicable to all athletes (such as older individuals, strength athletes, and women).

Additional comments

This is a relatively straightforward paper examining the effects of heavy resistance training and complex training on various running performance parameters in young male distance runners. My specific comments are in the attached annotated pdf. My major general comments are: 1) figures of the data are required, 2) speculative interpretations of results in the discussion section should be revised, 3) the discussion should include information on how these results could (should) be applied to female athletes, as well as what the implications are for future work, and 4) more clarity is needed in placing the study’s results within the broader context of the existing literature.

Annotated reviews are not available for download in order to protect the identity of reviewers who chose to remain anonymous.

---

## Round 0.2 · Minor Revisions

I have now received two external reviews of the revised manuscript and am happy to accept the article with one minor revision. As the first reviewer notes, the use of abbreviations provides an additional challenge to the reader to keep the variables straight. The manuscript would be much easier to read one abbreviation is used to refer to complex training throughout the manuscript, regardless of whether the text is referring to the method of training or the training group. Please make this one change and the resubmitted manuscript will be accepted.

Reviewer 1 ·

Basic reporting

The language is much improved! I commend the authors for taking all the reviewers' comments to heart and making substantial changes to this manuscript. The only thing I would suggest is that they shorten up the methods and results part of the abstract. I know I told them to expand, but I think they expanded a bit too much!

Experimental design

The new detail and clarifications to the methods are excellent

Validity of the findings

Love the addition of the figures and clarifications in the results including magnitude and direction of individual findings.

Additional comments

The improvements are great, excellent work!

Reviewer 2 ·

Basic reporting

Previously I raised the concern that the paper did not include figures demonstrating the major results. The authors now include Figures 2-5 which clearly show the results among training groups. In addition, they have edited the format of the in-text references and provided clarity in areas that were unclear.

Experimental design

The general basis for the study is sound, and the experimental design is clearly explained.

Validity of the findings

The authors have addressed my concerns regarding marking speculative comments as speculation and noting the caveat that the study’s results are not broadly applicable to all athletes such as older individuals and women.

Additional comments

The authors have satisfactorily addressed all of my previous comments. The new data figures are very useful for the reader to visualize the significant results. My only question regards the use of CT (complex training) versus CPX (complex training group) throughout the manuscript. It may make the manuscript easier to read if only one abbreviation is used to refer to complex training throughout the manuscript, regardless of whether the text is referring to the method of training or the training group.

---

## Round 0.3 · accepted · Accept

Thank you for the quick turn-around. We can now accept this for publication! Congratulations!